# Influence of Surface Modification of Titanium and Its Alloys for Medical Implants on Their Corrosion Behavior

**DOI:** 10.3390/ma15217556

**Published:** 2022-10-27

**Authors:** Łukasz Pawłowski, Magda Rościszewska, Beata Majkowska-Marzec, Magdalena Jażdżewska, Michał Bartmański, Andrzej Zieliński, Natalia Tybuszewska, Pamela Samsel

**Affiliations:** 1Department of Construction Materials, Institute of Manufacturing and Materials Technology, Faculty of Mechanical Engineering and Ship Technology, Gdańsk University of Technology, 80-233 Gdańsk, Poland; 2Department of Biomaterials Technology, Institute of Manufacturing and Materials Technology, Faculty of Mechanical Engineering and Ship Technology, Gdańsk University of Technology, 80-233 Gdańsk, Poland

**Keywords:** titanium, titanium oxide, carbon nanotubes, nano platinum, chitosan, polyamine, polymethacrylate, electrophoretic deposition, corrosion testing, microstructure

## Abstract

Titanium and its alloys are often used for long-term implants after their surface treatment. Such surface modification is usually performed to improve biological properties but seldom to increase corrosion resistance. This paper presents research results performed on such metallic materials modified by a variety of techniques: direct voltage anodic oxidation in the presence of fluorides, micro-arc oxidation (MAO), pulse laser treatment, deposition of chitosan, biodegradable Eudragit 100 and poly(4-vinylpyridine (P4VP), carbon nanotubes, nanoparticles of TiO_2_, and chitosan with Pt (nano Pt) and polymeric dispersant. The open circuit potential, corrosion current density, and potential values were determined by potentiodynamic technique, and microstructures of the surface layers and coatings were characterized by scanning electron microscopy. The results show that despite the applied modifications, the corrosion current density still appears in the region of very low values of some nA/cm^2^. However, almost all surface modifications, designed principally for the improvement of biological properties, negatively influence corrosion resistance. The reasons for observed effects can vary, such as imperfections and permeability of some coatings or accelerated degradation of biodegradable deposits in simulated body fluids during electrochemical testing. Despite that, all coatings can be accepted for biological applications, and such corrosion testing results are presumed not to be of major importance for their applications in medicine.

## 1. Introduction

Titanium and its alloys are well-known as resistant to general corrosion in many environments. Because of that and other outstanding features such as high biocompatibility, suitable mechanical properties except for wear resistance, and lightweight, they have become the best biomaterials for long-term endoprostheses or their components. However, they may appear prone to localized corrosion, wear corrosion, and general corrosion, in particular at low pH values and in saline-containing environments, i.e., just during the inflammation state [1,2,3].

There have been some papers in the last few years describing corrosion parameters of different coatings deposited or layers created on titanium [1,4,5,6,7,8,9,10,11,12,13,14,15] and its alloys: Ti6Al4V [16,17,18,19,20,21,22,23], Ti13Nb13Zr [14,24,25,26,27]. The coatings were designed and produced particularly to enhance corrosion resistance [1,4,6,7,8,9,11,16,17,18,19,21,22,23,24] or the corrosion characteristics were determined only to verify whether and to what extent the applied coating may affect the degradation process [5,10,12,13,14,15,20,25]. The main difference between coated materials was the presence or the absence of an artificial titanium oxide layer created by oxidation in different conditions [1,3,5,6,7,8,9,10,11,15,19,20] such as direct voltage anodic oxidation, micro-arc oxidation (MAO), or hydrothermal oxidation. The coatings other than rutile/anatase were composed of some metals [12], polymers [4,5,11,13,14,15,16], organic compounds [25], ceramics [5,9,10,23,24,25,26,27], oxides [5,6,8,9,12,19,22,24,25], graphene and carbides [1,18,20], and bioglass [17]. It is to underline that the most effective way to increase the corrosion resistance of titanium surfaces is to create a smooth and nonpermeable oxide layer. The different applied methods have some strengths and weaknesses. Low voltage anodic oxidation is a cheap and simple technique, capable of improving corrosion resistance and coating adhesion, but it needs techniques to measure and improve mechanical bonding and real-time/in-situ detection techniques to follow the course of surface reactions [28]. Regarding plasma electrolytic oxidation (micro-arc oxidation), the significant applications of this technique for Ti and its alloys include biocompatible, corrosion, and wear-resistant oxide coatings [29]. The method is strong because it produces porous Ca and P composite coatings containing adhesion-friendly osteocyte surfaces making it highly bioactive. On the other side, it is expensive, more difficult to design the coating properties, and still weakly commercialized. Finally, hydrothermal treatment is now relatively rarely proposed for titanium oxidation [7], but more recently was recommended for electronics [30] and water purification [31] than for implants. 

The corrosion environments vary: lysosome solution [4], NaCl solution [1,2,14,18,21], phosphate-buffered saline (PBS) [16], artificial saliva [7], Ringer’s solution [24] and SBFs of different compositions [5,6,8,11,12,14,19,23,25,26,27]. Table 1 shows a summary of these investigations. The values of corrosion potential or current have been determined in different corrosive liquids and after different mechanical surface treatments so that their highly diverse values are observed. This paper presents the results of corrosion studies performed by the authors on different coatings and substrates. The research has been conducted in five separate studies designated in the text by capital letters: -A: Testing of Ti after grinding, oxidation, surface decoration with Ag layer, and deposition of organic coatings composed of biopolymers (chitosan, Eudragit 100, and poly(4-vinylpyridine) (P4VP) organic amine);-B: Testing of Ti13Nb13Zr alloy obtained by an additive method and subjected to micro-arc oxidation in calcium-phosphorous with/without the addition of silver electrolytes;-C: Testing of Ti13Nb13Zr alloy after deposition layer-by-layer of carbon nanotubes and nanoparticles of titanium dioxide;-D: Testing of Ti, Ti6Al4V, and Ti13Nb13Zr alloys after their treatment with a pulse laser;-E: Testing of Ti after grinding and deposition of organic coatings composed of biopolymers (chitosan and polymeric dispersant Tween 20), and nanoPt.

The acquired results demonstrate that the effects of deposition of coatings are relatively high considering the differences in current density between coatings placed on extreme positions, the lowest and the highest values of corrosion current density. On the other side, even the least resistant coating can be considered highly protective. The sources of observed differences can be disparate, and they are discussed later.

**Table 1 materials-15-07556-t001:** Summary of recent results of corrosion tests published in the literature.

Coating	Substrate	Environment (Solution)	Corrosion Potential vs. SCE (V)	Current Density (µA/cm^2^)	Reference
None	Ti	SBF SBF SBF SBF SBF Hanks’s solution SBF SBF 0.6M NaCl SBF Lysosome -	−0.382 −0.167 −0.384 −0.770 −0.425 −0.611 −0.464 −0.76 −0.147 −0.228 −0.844 −0.23 ÷ −0.437	0.031 >0.1 0.272 1.45 1.60 4.57 0.19 178 30.7 0.038 4.78 0.10 ÷ 0.13	[8] [9] [6] [10] [11] [5] [13] [12] [14] [1] [4] [18]
None	Ti6Al4V	SBF SBF 3.5% NaCl SBF SBF	−0.583 0.374 0.210 −0.338 +0.074	0.24 62.6 0.078 2.53 2.37	[19] [21] [20] [16] [17]
None	Ti13Nb13Zr	Ringer’s solution SBF	−0.137 −0.487	0.276 0.052–0.078	[24] [26,27]
TiO_2_ (MAO)+ZrO_2_	Ti	SBF	+101 ÷ +222	0.012 ÷ 0.054	[8]
TiO_2_ (MAO)+ZnO TiO_2_ (MAO)+ZrO_2_ TiO_2_ (MAO)+ZnO+ZrO_2_	Ti6Al4V	SBF	+0.412 +0.246 +0.314	0.019 0.037 0.024	[19]
TiO_2_ (MAO)+GaO TiO_2_ (MAO)+Ta TiO_2_ (MAO)+GaO+Ta	Ti	SBF	−0.11 −0.05 −0.07	8.92 2.43 5.12	[12]
TiO_2_ (MAO)+boron carbide	Ti6Al4V	3.5% NaCl	−11 ÷ −480	0.09 ÷ 0.82	[20]
TiN (no heat treated) TiN (heat treated)	Ti	Artificial saliva	−276 ÷ +100	0.0003÷ 0.0139	[7]
TiN TiNO	Ti6Al4V	Artificial saliva	−0.32 −0.31 ÷ +0.42	0.34 0.06÷0.36	[22]
Graphene	Ti	-	−0.08 ÷ −0.19	0.04÷0.09	[18]
Graphene oxide (GO)	Ti	0.6M NaCl	−236 ÷ −289	0.015 ÷ 0.020	[1]
Hydroxyapatite (HAp)	Ti6Al4V	SBF	−0.503 ÷ −0.627	7.6 ÷ 18.0	[23]
HAp	Ti13Nb13Zr	SBF	−106	0.185	[24]
nanoHAp+nanoAg	Ti13Nb13Zr	SBF	−0.036 ÷ −0.482	0.007 ÷ 0.095	[27]
TiO_2_ (MAO)+HAp	Ti	SBF	−0.238 ÷ −0.401	0.044÷0.15	[10]
TiO_2_ (MAO)+(La)HAp	Ti	SBF	+0.238	>0.01	[9]
HAp Nb-HAp	Ti6Al4V	SBF	−0.307 +0.51	0.77 0.66	[21]
TiO_2_(NTs)+HAp+rGO/PCL TiO_2_(NTs)+brushite+rGO/PCL	Ti	SBF	−0.297 −0.597	0.37 0.82	[5]
HAp/ alginate GO/HAp/ alginate	Ti13Nb13Zr	SBF	−0.41 −0.33	24 1	[25]
Chitosan-P4VP-AgNPs	Ti	SBF	−0.371÷ −0.338	0.16 ÷ 0.19	[13]
Chitosan-Eudragit			−0.348	0.17	
Chitosan-Eudragit	Ti	SBF	−0.062 ÷ −0.340	0.06 ÷ 0.65	[15]
Chitosan-P4VP-AgNPs	Ti	SBF	−0.065 ÷ +0.068	0.86÷5.04	[14]
Chitosan	Ti	SBF	−0.309÷ −0.342	0.23÷0.45	[11]
Chitosan/gelatin Chitosan/gelatin/halloysite	Ti	Lysosome	−0.719 −1.598	7.90 374	[4]
UMHDPE Octadecylphosphonic acid (ODPA) UMHDPE/ODPA	Ti6Al4V	PBS	+0.074 −0.149 ÷+0.093 −0.055 ÷ +0.022	0.0015–0.007 0.058 ÷ 0.297 0.0000379	[16]
TiO_2_ (MAO)+silane TiO_2_ (MAO)+silane+GO	Ti	SBF	−67 −34	0.075 0.031	[6]
HAp+ZnO	Ti13Nb13Zr	SBF	−79 ÷ −197	0.083 ÷ 0.129	[24]
Bioglass (no or heat-treated)	Ti6Al4V	PBS	−0.307÷ −0.288	1.54/0.28	[17]
TiO_2_-chitosan/CuNPs	Ti13Nb13Zr		−0.217÷ −0.252	0.20 ÷ 0.30	[26]

MAO—micro-arc oxidation, rGO—reduced graphene oxide, SBF—simulated body fluid, PBS—phosphate-buffered saline.

## 2. Materials and Methods

### 2.1. Materials and Their Preparation

The substrates were titanium (Ti grade 2, EkspresStal, Lubon, Poland), Ti6Al4V (TIMET, Birmingham, UK), and Ti13Nb13Zr (SeaBird Materials Co., Baoji, China or Xi-an SAITE Metal Materials Development Co., Xi-an, China) alloys in solid form; only the last material was manufactured in Procedure B by the selective laser melting method (SLM 100, Realizer GmbH, Borchen, Germany) while using Ti13Nb13Zr powder (TLS Technik GmbH and Co. Spezialpulver KG, Bitterfeld-Wolfen, Germany). Different substrates have been used here because the presented results constitute parts of separate studies focused on Ti (dental implants), Ti6Al4V (hip joint implants), and Ti13Nb13Zr alloy (anticipated to be used in increasingly more applications in implantology as a non-toxic material). Based on several previous studies, no greater differences in corrosion rate should appear for dissimilar substrates, with titanium being the most resistant material. The use in procedure D of all three alloys was justified by the verification of the effect of the material preliminary on the microstructure of the substrate layer formed by laser treatment.

The samples were discs of an area of 2–7 cm^2^ and a thickness of 3–4 mm. The mechanical surface treatment was performed in each case by wet grinding (Saphir 330, ATM GmbH, Mammelzen, Germany) to a final gradation of #800 with SiC sandpapers (Struers Company, Cracow, Poland), followed by cleaning with acetone, isopropanol (all fluids chemically pure, delivered by POCH S.A., Gliwice, Poland) and distilled water (HLP 5, Hydrolab, Straszyn, Poland) for 2–10 min in each solvent. In Procedure D, etching in 3% HF was performed. After ultrasonic cleaning, the specimens were dried in ambient air. 

Laser treatment was applied in Procedure D using a short pulse Nd:YAG laser (Trulaser Station 5004, TRUMPF, Dizingen, Germany) at an average power of laser beam 4500 W, beam pulse power 100 W, time speed of laser beam 1 ms, frequency 25 Hz, and overlapping 50%. Argon 5.0 (Linde Gaz Poland Sp., Kraków, Poland) was used to reduce surface oxidation during laser treatment. 

### 2.2. Electrochemical Oxidation

Electrochemical anodic oxidation (AO) in Procedure A was carried out in an electrolyte containing 10 mL of 85% orthophosphoric acid (H_3_PO_4_, Sigma Aldrich, St. Louis, MO, USA) and 1.2 mL of 40% hydrofluoric acid (HF, POCH, Gliwice, Poland) per 150 mL of distilled water. Two electrodes, a titanium sample as the anode and a platinum mesh electrode as the cathode, were connected to a direct current source (MCP Corp., Shanghai, China). The distance between the electrodes was set at 10 mm. In Procedure A, according to previous studies [15], the anodization voltage and time were adjusted to 20 V and 20 min, respectively. After oxidation, the samples were washed thoroughly with distilled water.

The micro-arc oxidation (MAO) was conducted in Procedure B while using a DC power supply (MR100020, B&K Precision Corp., Yorba Linda, CA, USA) according to the procedure detailed in [32,33], under two voltages of 300 and 400 V, a constant current of 216 A/m^2^ for 15 min in the electrolyte contained 0.1 M of calcium glycerophosphate C_3_H_7_CaO_6_P (GP), 0.15 M of calcium acetate Ca(CH_3_COO)_2_ (CA), with and without the addition of 0.006 M silver nitrate AgNO_3_ (all fluids chemically pure, delivered by POCH S.A., Gliwice, Poland). After the MAO process, the samples were washed in ultrapure water and dried.

### 2.3. Deposition of Coatings

The coatings were deposited in Procedures A, C, and E. 

In Procedure A, six surface modifications were applied, as shown in Table 2. Procedure A4 was based on grinding the titanium samples, followed by electrochemical oxidation and successive decoration with silver nanoparticles by electro-reduction of silver nitrate (AgNO_3_, VWR Chemicals, Leuven, Belgium) according to the procedure detailed in [34]. The silver deposition was carried out from a solution containing 0.005 g of AgNO_3_ in 1 L of distilled water using a potentiostat (Atlas 0531, Atlas Sollich, Gdansk, Poland) for a voltage of −1.2 V for only 15 s. After deposition, the sample was likewise washed with distilled water. The deposition (A5) was carried out from suspensions containing 0.1 g of chitosan (purity > 99%, MW 310–375 kDa, degree of deacetylation > 75%, Sigma-Aldrich, St. Louis, MO, USA) with the addition of 0.25 g of Eudragit E 100 (purity 99.9%, MW 47 kDa, Evonik Industries, Darmstadt, Germany) per 100 mL of 1% acetic acid (Stanlab, Gliwice, Poland). The deposition procedure was carried out according to the protocol described earlier [13,14], applying voltages of 10 V for 1 min at room temperature. The deposited coatings were rinsed gently with distilled water afterward. The modification A6 also involved depositing coatings on a titanium sample after grinding, electrochemical oxidation, and silver decoration, except that an equal amount of poly(4-vinylpyridine) (MW~160,000; Sigma-Aldrich, St. Louis, MO, USA) was introduced instead of Eudragit E 100.

In Procedure C, the deposition of carbon nanotubes (CNTs) was performed based on the CNTs suspension made of 0.1 g of polyhedral functionalized CNTs (powder) and 40 mL of distilled water. The substrates were sonicated in an ultrasonic bath for 60 min. The CNTs concentration in the suspension was 0.25 wt%. The electrophoretic deposition process was carried out using the SPN110-01C power supply (Multi-Com, Kolbuszowa, Poland), two electrodes, and a beaker with a reactive suspension. The titanium sample played the role of the cathode (+), and the steel electrode the anode (−). The time and voltage were constant for each sample, respectively, 30 s and 20 V. The coatings thus obtained were allowed to dry for 24 h at room temperature in the air [35].

In the same procedure, the deposition process of titanium oxide nanoparticles was started after 24 h from the deposition of MWCNTs, i.e., the time necessary for the coating to dry. The day before depositing the titanium oxide nanoparticles, 50 mL of isopropanol and 0.5 mL of Tween 20 polymeric dispersant were poured into each of the two beakers. The amount of titanium oxide nanoparticles in one suspension was 0.15 g, and in the other, 0.30 g of rutile powder. The measured substrates were sonicated for 4 h in an ultrasonic bath, and on the day of settling, for an additional 2 h. The electrophoretic system was the same as for the CNTs deposition. The only difference was the method of connecting the electrodes—the titanium sample was the anode (−), and the steel electrode was the cathode (+). The deposition time was 4 min. The variable parameter was the voltage, the value of which was 50 and 60 V according to [36]. 

In Procedure E, the electrolyte was prepared by dispersing 1 g of high-weight chitosan (degree of deacetylation > 75%; Sigma Aldrich, St. Louis, MO, United States) in 1 L of 1% acetic acid (Polskie Odczynniki Chemiczne, Gliwice, Poland) following previous similar research [27]. The electrolyte was homogenized using the magnetic stirrer at 250 rpm for 24 h at room temperature. Then three different suspensions based on the prepared electrolyte were used: without or with 1 mL of Tween 20 (Polysorbate 20) (Sigma Aldrich, St. Louis, MO, United States) diluted in 1 L of 1% (*v*/*v*) acetic acid, and with 1 mL of Tween 20 and 0.05 g of platinum nanoparticles (nanoPt; average powder grain about 50 nm; purity 99.99%, MKNano, Mississauga, Canada). The Tween 20 and Pt nanoparticles were added 1 h before deposition and then homogenized using the magnetic stirrer at 250 rpm. The Ti13Nb13Zr specimen was a cathode, and platinum was an anode. The electrodes were placed at a distance of 10 mm. The DC power source (MCP/SPN110-01C, Shanghai MCP Corp., Shanghai, China) was applied. The EPD was performed at 20 V for 1 min at room temperature. Finally, the as-deposited composite coatings were rinsed with distilled water and air-dried at room temperature for 48 h.

The details of the applied Procedures are listed in Table 2.

### 2.4. Characterization of Microstructure

The microstructures of the samples’ surfaces after modifications were examined with scanning electron microscopy (SEM, FEI Quanta FEG 250, Hillsboro, OR, USA and JSM-7610F, JEOL Europe SAS, Croissy-sur-Seine, Yvelines, France). Before imaging, the coated samples were sputtered whenever necessary with a gold layer of 10 nm thickness using DC magnetron sputtering (EM SCD 500, Leica, Vienna, Austria) in Procedure A and a 10 nm layer of chromium in Procedure E for both in pure Ar plasma conditions.

### 2.5. Corrosion Tests

Corrosion tests on samples after the proposed modifications were carried out with a potentiostat/galvanostat (Atlas 0531, Atlas Sollich, Gdansk, Poland) using a three-electrode system, together with the control software paired with the AtlasCorr05 software directly and independently calculating the corrosion parameters (corrosion current density and corrosion potential) based on Tafel extrapolation and drawing the polarization curves. The working electrode was the test sample, the counter-electrode was a platinum rod, and a saturated calomel electrode (SCE) was employed as the reference electrode. During the test, all electrodes were immersed in several simulating human body fluids (SBF) with the chemical compositions shown in Table 3 for applied procedures. The use of different solutions has been justified by the anticipated application of surface-modified implants. Therefore, the corrosion studies in procedures A and E were performed in artificial saliva (typical for dental implants), procedure B in the most corrosive physiological salt (suitable for all implants), and corrosion investigations in C and D were carried out in the Ringer’s solution, used as the test fluid for many implants. The magnetic stirrer with a heating plate, maintaining a constant temperature of 37 °C and making 100 rpm, preserved a homogeneous temperature of the solution throughout the volume.

Firstly, the open circuit potential (OCP) values were determined. Then the corrosion curves were recorded using the potentiodynamic method. The measurement was carried out for potential ranging from about −1.0 to +1.0 V, at a potential change rate of 1 mV/s, often applied in similar tests on titanium. Using Tafel extrapolation, the values of corrosion potential (E_corr_) and corrosion current density (j_corr_) were determined with the special software AtlasLab (Atlas-Sollich, Rębiechowo, Poland), attached to the potentiostat. The collected data were statistically analyzed using OriginPro software (8.5.0 SR1, OriginLab Corporation, Northampton, MA, USA). The normal distributions of the data were evaluated using the Shapiro–Wilk test. The results of the measured values of corrosion parameters are presented as mean ± standard deviation (SD). Statistical analysis was conducted by one-way ANOVA with Bonferroni correction with statistical significance set at *p* < 0.05.

## 3. Results and Discussion

The surface views for all tested specimens are shown in Figure 1, Figure 2, Figure 3, Figure 4 and Figure 5. In Figure 1, the nanotubular surfaces and slightly uniform layers are observed (1b and 1c) as the effects of anodic oxidation in the presence of fluorides and further silver precipitation. The presence of nanosilver atoms/particles is seen in Figure 1c as a non-continuous white layer. The chitosan-based composite coatings are continuous (Figure 1d,e), and the addition of P4VP instead of Eudragit makes the surface a little rougher. The chemical phase compositions of each examined coating were already characterized by XRD, XPS, EDS, and Raman spectra for chitosan/Eudragit/Ag coatings [13,38] and by EDS and FTIR examinations for chitosan/P4VP/Ag coatings [14]. These earlier results showed that the nanotubular oxide formed an anatase phase, Ag appeared in the form of elementary nanoparticles, and both chitosan and P4VP were proved to form coatings.

In Figure 2, the surfaces show a typical MAO appearance with many characteristic pores. The chemical and phase compositions of tested coatings have already been examined with the EDS and XRD [33]. The Ti, O, C, and P elements were recorded as what could be interpreted as the presence of an oxide layer, identified as rutile and anatase. No phase containing Ca or P in the XRD study can be explained by either an absence of any crystalline phase such as apatite, a possible appearance of the amorphous phase of phosphate, or an appearance of a crystalline phase in the amount below the detection limit. 

**Figure 2 materials-15-07556-f002:**
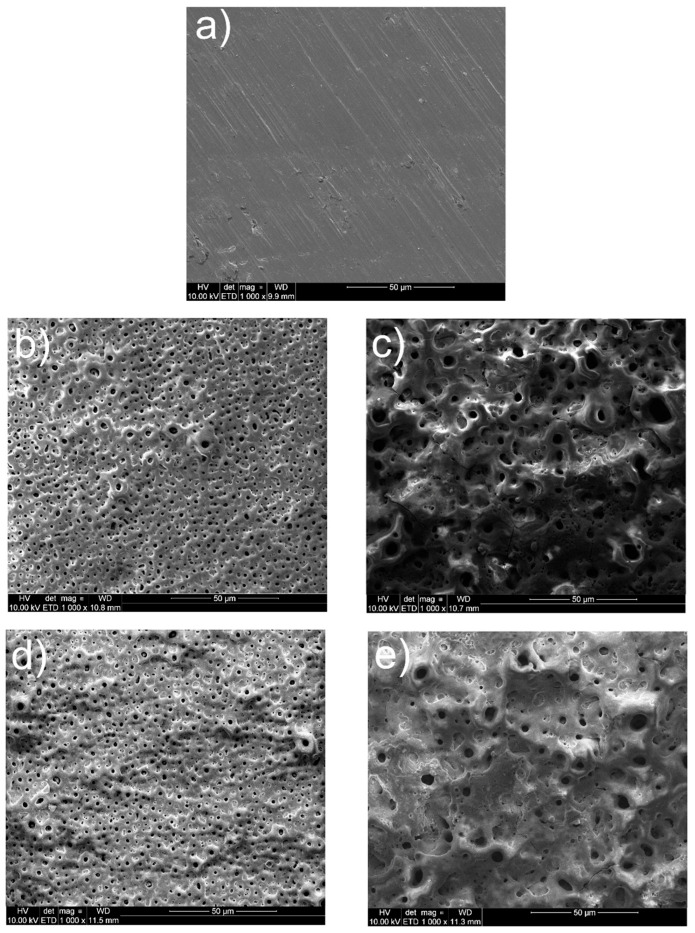
Microstructures of surfaces after different modifications: (**a**) ground, (**b**) ground and subjected to MAO at 300 V in CaP + GP solution, (**c**) ground and subjected to MAO at 400 V in CaP + GP solution, (**d**) ground and subjected to MAO at 300 V in CaP + GP + Ag solution, (**e**) ground and subjected to MAO at 400 V in CaP + GP + Ag solution.

In Figure 3, highly uneven coating surfaces are visible. The EDS examinations [39,40] showed that the EDS spectra showed a presence of carbon, copper, titanium, and oxygen, confirming in such a way the formation of MWCNTs and TiO_2_ on the surface. 

**Figure 3 materials-15-07556-f003:**
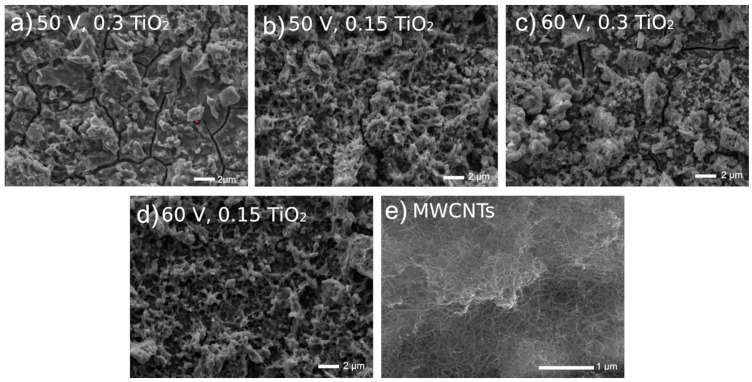
Microstructures of Ti13Nb13Zr surfaces previously ground after being coated with MWCNTs and then coated with a nanotubular titanium oxide layer at different voltages and oxide contents in a bath.

In Figure 4, the surfaces look relatively even, with a wavy image, and several tracks after a passage of the laser beam are observed. At this research stage, neither EDS nor XRD was performed, and they are planned for the future.

**Figure 4 materials-15-07556-f004:**
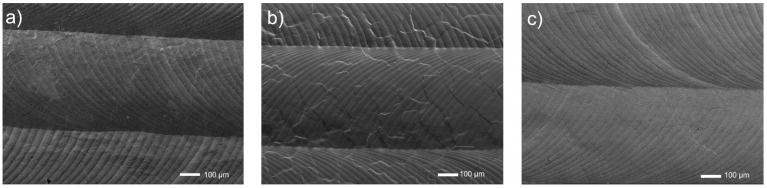
Microstructures of surfaces of titanium and its alloys, ground, and laser modified: (**a**) Ti, **(****b**) Ti6Al4V, (**c**) Ti13Nb13Zr.

Finally, the coated surfaces prepared according to the last procedure E (Figure 5) are breathy for deposition without Tween 20, and even much smoother when the polymeric dispersant has been added. The chemical examinations were already made by EDS for a similar composition but without Pt (such studies are planned to be carried out in the near future). The presence of chitosan after deposition was already proved by the FTIR and Raman studies [41].

**Figure 5 materials-15-07556-f005:**
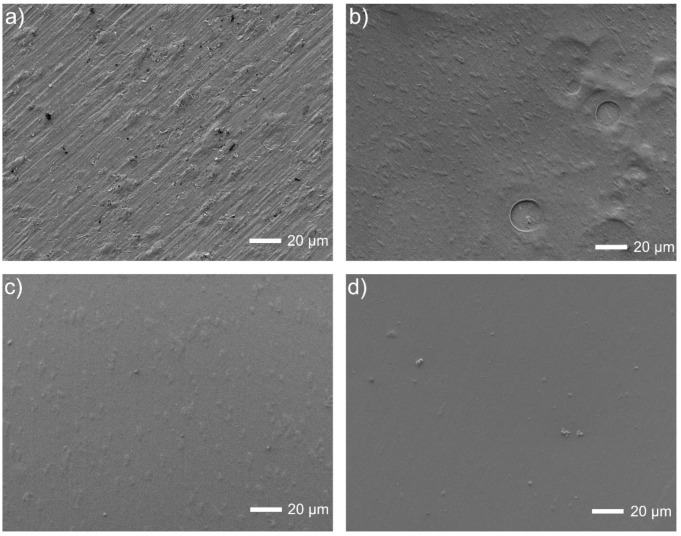
Microstructures of Ti13Nb13Zr surfaces: (**a**) ground, (**b**) ground and chitosan-coated, (**c**) ground and chitosan/Tween 20 coated, (**d**) ground and chitosan/Tween 20/nanoPt coated.

The results of the corrosion tests are illustrated in Figure 6, Figure 7, Figure 8, Figure 9 and Figure 10 and Table 4. Figure 6 shows the relatively low change in OPC values during even 3500 s of exposure. The most pronounced increase in OCP was noticed for both chitosan-based coatings, presumably because of a slight deposition of phosphate from a corrosive solution (artificial saliva). The lowest corrosion rates and the highest corrosion potential values for these two samples prove the protective property of both chitosan-based coatings. The high corrosion rate for a Ag decorated sample is due to only partial coverage of a sample and perhaps an appearance of the electrochemical cell Ti6Al4V-Ag. The run of curves shows that only AO alone improves the corrosion resistance described by a shift of E_corr_ into more negative values. On the contrary, the addition of Eudragit to the chitosan-based coating moves the corrosion potential into a more positive value. As all curves are similar to each other, we might say that all applied surface treatments have changed the ohmic resistance. Thus, the biodegradable chitosan-based coatings are prone to chemical degradation during the testing, revealing the bare surface (not oxidized). 

In Figure 7, the OCP values are all positive; both changes in voltage value and the presence or absence of silver have no clear and understandable effects on OCP. More information can be gathered by looking at Table 1. All surface treatments have negative effects on corrosion current density, i.e., on corrosion rate. The effect of silver presence seems weak and undefined. The worsening of corrosion behavior for all surface treatments is evidence that any inhomogeneity created by deposits of silver likely results in moderate corrosion. 

The application of MAO is a well-known process that usually causes an increase in corrosion resistance, even if its main aim is to build bioactivity. The higher applied voltage had a negative effect on the corrosion resistance. The lower voltage is presumably more promising for also creating an anticorrosion layer. Such results are not surprising, as confirmed by several works [7,10,17]. As can be noticed, the MAO coatings built on porous SLM-made structures are characterized by the worst corrosion resistance among all of the presented modifications. The highly porous surface with visible microcracks attributed to thermal stresses caused by rapid solidification of the molten oxide allows the corrosive fluid to penetrate and react with the substrate. Moreover, it is worth noticing that the poor corrosion resistance is strictly related to the manufacturing method of the substrate. Due to the possible differences in grain boundary density and rate of α and β phases, the microstructures of selective laser-melted specimens are generally more susceptible to corrosive attack in comparison to solid material [42]. An additional reason could be insufficient oxidation of a surface inside the deep pores followed by an appearance of a difference in potentials and an electrochemical cell ‘surface top—pore bottom’.

In Figure 8, no significant change in OCP is observed, which is evidence of the coatings’ stabilities. A slight decrease in potential can be observed in coatings containing less titania, perhaps not so stable in a used solution. The runs of potentiodynamic polarization curves indicate influencing the behavior by both resistance and activation depolarization processes. The exceptionally high corrosion rate observed for the coating with less titania and deposited at higher voltage can be attributed to the formation of coating with mainly CNTs, which are not as protective as titanium oxide, and also to the likely presence of a more porous coating at this high voltage. Next, the MWCNTs + TiO_2_ bi-layer coatings demonstrate that without a titania outer layer, they are highly susceptible to corrosion. The cause can be a weak adherence of the inner layer to the surface and, even more importantly, non-uniform CNTs coating with probably numerous free spaces between tubes. Thus, such an outer layer gives protection only at lower titanium dioxide content and at lower voltage values. It seems that the microstructure of the coating is highly sensitive to the process parameters or even unstable, and this material system needs further investigation. There are no such results obtained for Ti or its alloys, but in [43], the corrosion rate in the presence of the MWCNTs coating was found to be 10-fold lower (0.0966 mm/year) compared to the uncoated porous Ti-30%Ta alloy. The other results [44] showed a remarkable increase in corrosion resistance for the MWCNTs-hydroxyapatite bi-layer. 

**Figure 7 materials-15-07556-f007:**
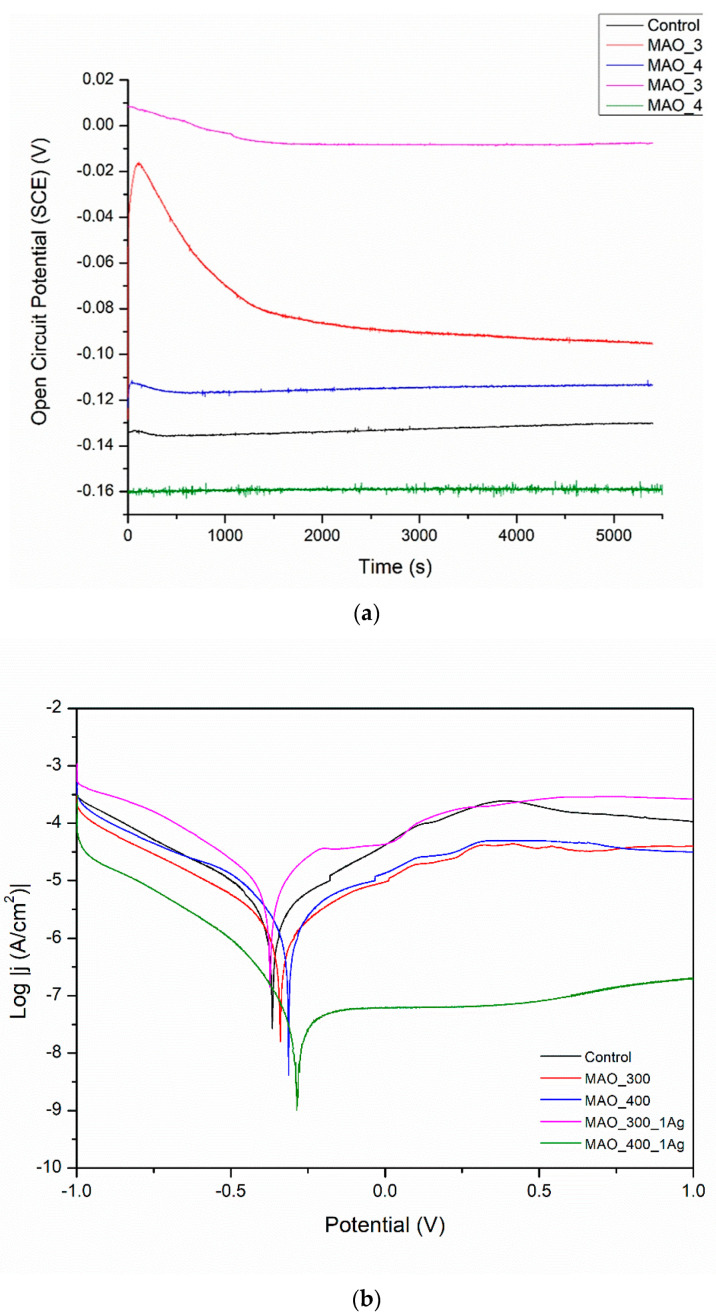
(**a**) Open circuit potential for different modifications, (**b**) potentiodynamic corrosion curves obtained for Ti13Nb13Zr substrate after its surface modifications. Control—ground, MAO_300—ground, and MAO treated at 300 V in CaP + GP solution, MAO_400—ground, and MAO treated at 400 V in CaP + GP solution; MAO_300_1Ag—ground, and MAO treated at 300 V in CaP + GP + Ag, MAO_400_1Ag—solution ground, and MAO treated at 400 V in CaP + GP + Ag solution.

**Figure 8 materials-15-07556-f008:**
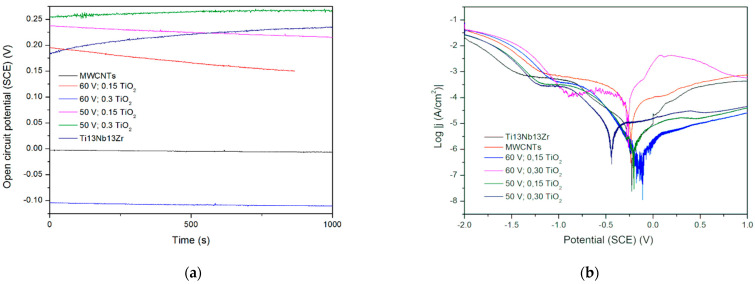
(**a**) Open circuit potential for different modifications, (**b**) potentiodynamic corrosion curves obtained for Ti13Nb13Zr substrate after its surface modifications. Ti13Nb13Zr—ground, MWCNTs—ground and coated in MWCNTs solution, other curves—ground, coated in MWCNTs solution and coated again in TiO_2_ solution (layer-by-layer), at different contents and voltages.

Figure 9 demonstrates the behavior of three different titanium biomaterials subjected to preliminary laser treatment. The change in activation depolarization seems to be the reason for the change in corrosion current. On the other side, the laser modification always makes the corrosion resistance worse, probably due to the appearance of the voltage difference between the contact area of successive remelted zones and the central parts of the zones. The effect of the material composition is observed, with the Ti6Al4V alloy being the most resistant. However, this difference can be rather attributed to the change in microstructure after laser treatment; the differences in corrosion rate for no-treated materials appear relatively small. Finally, the laser treatment highly negatively affected the corrosion resistance, likely by an occurrence of excessive roughness and higher phase and chemical non-uniformity of the surface. Therefore, it should not be recommended as the mechanical treatment of titanium implants. However, the other results showed that either laser treatment did not change corrosion behavior [45] or laser-textured surfaces had a good effect on titanium corrosion resistance in NaCl solution [46]. 

**Figure 9 materials-15-07556-f009:**
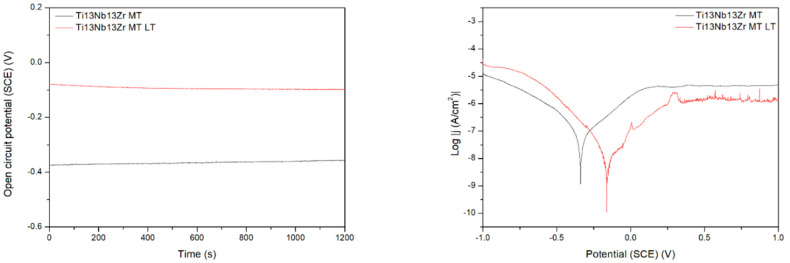
Open circuit potential (**a**) and potentiodynamic corrosion curves (**b**) obtained for Ti (bottom), Ti6Al4V (middle), and Ti13Nb13Zr (top), as ground (MT) and laser treated (LT) substrates.

Finally, the effects observed in Procedure E can be easily explained. The deposition of chitosan results in the deterioration of corrosion behavior as chitosan forms a relatively soft and porous deposit. The addition of nano platinum makes the corrosion resistance better than for pure chitosan, presumably because of the passive character of platinum and blocking by nanoparticles of the formation of possible corrosion tunnels. 

**Figure 10 materials-15-07556-f010:**
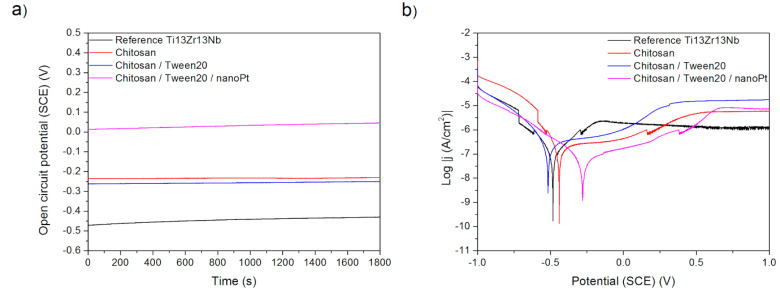
(**a**) Open-circuit potential vs. time and (**b**) potentiodynamic corrosion curves obtained for Ti13Nb13Zr substrate after its surface modifications: Reference—ground, Chitosan—ground and coated with chitosan, Chitosan/Tween 20—ground and coated with chitosan and Tween 20 solution, Chitosan/Tween 20/nanoPt—ground and coated with chitosan/Tween 20 + nanoPt solution.

The results show that despite the applied modification, the corrosion current density still appears in the region of low values, from about single nA/cm^2^, often of tens or more often hundreds, but after MAO, even a thousand or more. It is important to remember that whatever surface modification is applied, it is designed mainly to improve biological behavior, i.e., maintain or increase biocompatibility (sufficient for titanium and its alloys without any further surface treatment), enhance bioactivity (usually not existing or weak for these biomaterials), and sometimes, more and more often, introduce important antibacterial protection, at least during the first 3–30 days after surgery. On the other side, the used surface treatment cannot be cytotoxic and well-bonded to the titanium substrate, at least before it is exposed to body fluids. The observed discrepancies can be related, to an undefined extent, except different SBF and coatings or layers, in addition to factors such as moderate reliability of determination of corrosion parameters based on Tafel plots for titanium and its alloys and a real surface different from the geometrical surface because of surface roughness or porosity. In particular, the Tafel plots, even calculated by the software, are not as linear as they should be here for precise calculation, and the tests might be made under conditions at which the Tafel extrapolation is more or less erroneous [47]. In addition, for the surface prepared by MAO, significant porosity is obtained because of the many pores, resulting in a serious overestimation of corrosion current density. The coatings can also be more or less permeable for test solutions, which means the real surface exposed to the test solution can be lower than a geometrical one. Thus, the corrosion current densities shown here can be far from real corrosion rate values and can be used only to compare the effects of a specific coating.

The corrosion potentials determined by the software and OCP potentials measured at the beginning of testing are a little diverse. That means that the Tafel extrapolation has not been perfect. Therefore, we do not consider here the OCP values as any helpful measure of corrosion tendency. 

The most important results obtained for different layers and coatings are, according to the literature (Table 1), comprised of even values as unbelievably low as 0.0003 to 374 nA/cm^2^, because of possible errors and reasons discussed above. Therefore, we decided to calculate the corrosion protection efficiency Z according to the well-known equation from corrosion science:

Z = (i_c_ − i_s_)/i_s_
(1)

where i_c_ is the corrosion current density of the coated sample, and i_s_ is the corrosion current density of the uncoated sample (substrate: Ti or its alloy). 

It is often assumed that surface modification should enhance corrosion resistance together with important biological properties such as biocompatibility and bioactivity. Following this, the negative protection efficiency observed for a majority of procedures (Table 5) might be a base, from the first view, to reject such material systems. Fortunately, it is not true: coatings that are very stable and fully resistant to flowing body fluids, including the most aggressive NaCl solution, cannot be active to the osteoblasts. The ideal situation is when the coating or layer is permanently bonded to the implant and allows bone cells to adhere, migrate and grow on the surface. However, because of stresses acting on the implant during human activities, followed by possible cracking and degradation of the coating, and also because of more and more considered long-term antibacterial protection by, e.g., antibiotics or nanosilver placed inside slowly biodegradable surface deposits, the coating should stepwise transform in 3-6 months to a human apatite by a chemical reaction between a coating and a bone tissue. Despite that, the corrosion current densities above a certain limit and negative values of anticorrosion protection efficiency higher than another limit should be rejected. We think that such a limit might be set up at no more than 100 nA/cm^2^ and corrosion protection effectivity at −100% or higher. That means that among all tested surface modifications, those accepted include coatings: AO + Ag layer + chit/EU layer; CNTs layer + TiO_2_ layer (0.15 g, 50 V); chitosan/Tween 20/nanoPt coating or chitosan alone. This conclusion is based on the corrosion protection efficiency from Table 5. However, when taking into account the observed current density values, much more modifications are suitable, i.e., a majority of coatings based on MWCNTs and TiO_2_, and a laser treatment for Ti6Al4V alloy could also be accepted. However, as said before, the corrosion current density value has an important limitation: its value depends on the type of corrosive fluid. 

The temporary decrease in corrosion rate manifests by increasing corrosion current density and shift of corrosion potential to more positive values (from negative to positive directions). However, those values are scarcely affected by surface modification and can be neglected.

The observed differences can be due to various reasons. Some have already been considered, but the desired and designed biodegradation is also important. At first, two types of coatings, both based on chitosan (Procedures A and E), are biodegradable at lower values of pH (in acidic solutions). It is a reason to consider them as the main substrates for antibacterial smart pH-related biodegradable coatings. The corrosion testing in the cathodic area creates the alkaline solution, and in the anodic area—the acidic solution. Thus, the coatings change during the testing, and their measurements cannot be helpful in accurately describing their corrosion behavior in neutral conditions. Therefore, for such coatings, the other techniques, such as impedance spectroscopy or the seemingly forgotten linear polarization method, seem more reasonable, and the potentiodynamic method should be limited to highly stable coatings and metals. 

In these tests, almost all surface modifications, designed for the improvement of biological and mechanical properties, negatively influenced the corrosion resistance. The reasons for observed effects can be various: formation of pores and unevenness followed by an increased permeability in multicomponent structures and fast degradation of coatings in simulated body fluids accelerated by cathodic or anodic polarization during testing. Despite that, all coatings can be accepted for biological applications, and such corrosion testing results are presumed not to be the reason for the rejection of less resistant coatings, but the corrosion factor should be taken into account anyway when choosing the best compositions and deposition parameters.

In summary, it can be concluded that the best anticorrosion properties possess ceramic or ceramic-based coatings, and not polymer or polymer-based ones, in SBF or saline environments. Therefore, if the coating has a significantly negative effect on corrosion behavior, it is suitable to add to it some ceramic components. On the other side, the major role of coatings is not to protect titanium from corrosion, and often, even worsening corrosion resistance can be accepted if bioactivity or antibacterial behavior is achieved. In addition, the obtained corrosion testing results are dependent on test parameters and, on the other hand, on the already mentioned coating thickness and compactness but also uniformity, toughness, the appearance of cracks or not, and many other factors. 

## 4. Conclusions

The applied surface modifications of titanium and its alloys destined to increase in bioactivity can deteriorate corrosion resistance. The decrease in corrosion resistance manifests by increasing corrosion current density and shift of corrosion potential to more positive values. Despite that, the corrosion current density, being the reasonable index of corrosion rate, remains at low values of some nA/cm^2^, which is fully satisfactory. In the opposite case in which the corrosion current density related to a geometrical surface exceeds 100 nA/cm^2^, the layers or coatings need to be improved.

The observed differences in corrosion behavior can be related to various features and processes. Most importantly, the coatings based on chitosan are biodegradable at lower values of pH and may change their structure and thickness due to local acidification at anodic polarization. It is postulated that, for such coatings, the impedance spectroscopy and the linear polarization method will be more suitable than the potentiodynamic method. 

The MWCNTs + TiO_2_ bilayer coatings demonstrate moderate corrosion dissolution, which is particularly observed at low titania content and low potential values suggesting that the microstructure of the coating is highly sensitive to the process parameters or even unstable.

The composite oxide-based layers created by the MAO on porous titanium surfaces are the worst among the coatings investigated here due to many possible factors, with the appearance of an electrochemical cell and the difference in potentials between the oxide surface and non-oxidized pore bottom.

The laser treatment highly negatively affected the corrosion resistance due to excessive roughness and phase and chemical surface non-uniformity. 

The best anticorrosion properties can be achieved by ceramic or composite ceramic-based coatings and not by polymer or polymer-based layers. 

The most important role of bioactive coatings is to enhance bioactivity at no cytotoxicity, even if such modification involves a moderate corrosion increase. 

## Figures and Tables

**Figure 1 materials-15-07556-f001:**
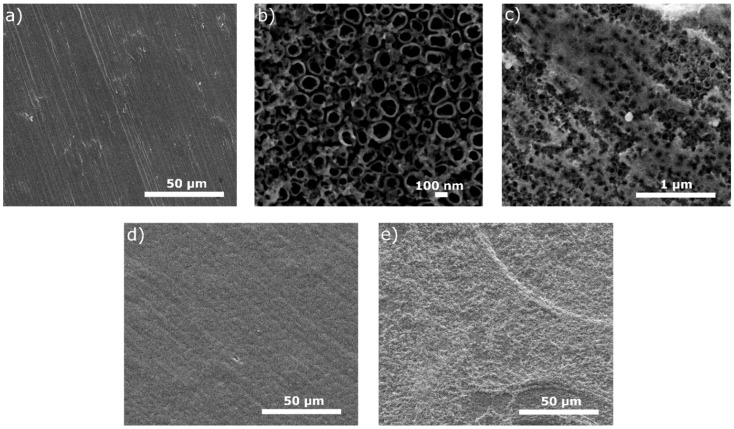
Microstructures of surfaces after different modifications: (**a**) ground, (**b**) ground and anodically oxidized to obtain titanium nanotubes, (**c**) ground, oxidized, and coated with Ag layer, (**d**) ground, oxidized, Ag-decorated and coated with chitosan/Eudragit 100, (**e**) ground, oxidized, Ag-decorated and coated with chitosan/P4VP.

**Figure 6 materials-15-07556-f006:**
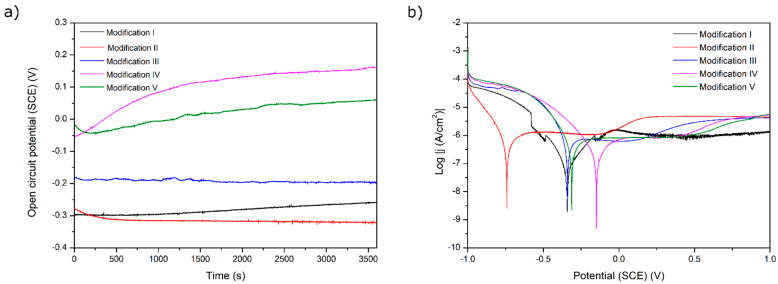
(**a**) Open-circuit potential vs. time and (**b**) potentiodynamic corrosion curves obtained for Ti substrate after its surface modifications: I—ground, II—anodically oxidized to obtain titanium nanotubes, III—coated with Ag layer, IV—ground, oxidized, Ag decorated and coated with chitosan/Eudragit 100, V—ground, oxidized, Ag decorated and coated with chitosan/P4VP.

**Table 2 materials-15-07556-t002:** Details of surface modifications applied in various procedures.

Procedure	Substrate	Mechanical Surface Treatment	Electrochemical Oxidation/Laser Treatment Parameters	Coating Deposition Parameters
A1 A2 A3 A4 A5	Ti	Sandpaper No. 800, the last	None	None
		As above	AO; H_3_PO_4_ + HF; 20 V; 20 min	None
		As above	None	0.005 g Ag; −1.2 V; 15 s
		As above	AO; H_3_PO_4_ + HF; 20 V; 20 min	0.005 g Ag; −1.2 V; 15 s 0.1 g chitosan + 0.25 g Eudragit; 10 V; 1 min
		As above	As above	0.005 g Ag; −1.2 V; 15 s 0.1 g chitosan + 0.25 g P4VP; 10 V; 1 min
B1 B2 B3 B4 B5	Ti13Nb13Zr	As above	None MAO; 0.15M Ca, O.1M GP; 300 V; 15 min MAO; 0.15M Ca, O.1M GP; 400 V; 15 min MAO; 0.15M Ca, O.1M GP, 0.006M AgNO_3_; 300 V; 15 min MAO; 0.15M Ca, O.1M GP, 0.006M AgNO_3_; 400 V; 15 min	None
C1 C2 C3 C4 C5 C6	Ti13Nb13Zr	As above	None	None 0.25% MWCNTs; 20 V; 30 s +0.15 g nanoTiO_2_; 50 V; 4 min 0.25% MWCNTs; 20 V; 30 s +0.15 g nanoTiO_2_; 60 V; 4 min 0.25% MWCNTs; 20 V; 30 s +0.30 g nanoTiO_2_; 50 V; 4 min 0.25% MWCNTs; 20 V; 30 s +0.30 g nanoTiO_2_; 60 V; 4 min 0.25% MWCNTs; 20 V; 30 s
D1 D2 D3 D4 D5 D6	Ti Ti6Al4V Ti13Nb13Zr Ti Ti6Al4V Ti13Nb13Zr	As above	None None None Laser power beam 4500 W, pulse power 100 W, time speed of laser beam 1 ms, frequency 25 Hz	None
E1 E2 E3 E4	Ti13Nb13Zr	As above	None	None 0.1% chitosan; 20 V; 1 min 0.1% chitosan + 0.01% Tween 20; 20 V; 1 min 0.1% chitosan + 0.01% Tween 20 +0.005% nanoPt; 20 V; 1 min

**Table 3 materials-15-07556-t003:** Chemical compositions of simulated body fluids used in various procedures.

Procedure Designation	Composition (g/L)
A and E	Artificial saliva: (NH_4_)_2_CO, 0.13; NaCl, 0.7; NaHCO_3_, 1.5; Na_2_HPO_4_, 0.26; K_2_HPO_4_, 0.2; KSCN, 0.33; KCl, 1.2; by [37]
B	Physiological salt: NaCl, 9
C and D	Ringer’s solution: NaCl, 8.6; CaCl_2_, 0.48; KCl, 0.30

**Table 4 materials-15-07556-t004:** Results of corrosion tests.

Procedure Designation and Substrate	Substrate	Surface Treatment	E_corr_ (V(SCE))	i_corr_ (nA/cm^2^)
A1 A2 A3 A4 A5	Ti	Only MT	−0.341 ± 0.04	152 ± 109
		No MT; AO	−0.533 ± 0.01	1112 ± 81
		MT; Ag layer	−0.338 ± 0.03	810 ± 87
		MT; AO + Ag layer + chit/Eudragit layer	−0.113 ± 0.01	303 ± 45
		MT; AO + Ag layer + chit/P4VP layer	−0.293 ± 0.01	788 ± 46
B1 B2 B3 B4 B5	Ti13Nb13Zr	Only MT	−0.325 ± 0.03	2647 ± 720
		MT; MAO Ca + GP; 300 V	−0.341 ± 0.01	4573 ± 251
		MT; MAO Ca + GP; 400 V	−0.316 ± 0.03	8500 ± 3024
		MT; MAO Ca + GP + Ag; 300 V	−0.333 ± 0.01	5483 ± 2505
		MT; MAO Ca + GP + Ag; 400 V	−0.331 ± 0.03	7733 ± 3858
C1 C2 C3 C4 C5 C6	Ti13Nb13Zr	Only MT	−0.225 ± 0.02	4.22 ± 0.2
		MT; CNTs layer + TiO2 layer (0.15 g, 50 V)	−0.169 ± 0.01	1.4 ± 0.3
		MT; CNTs layer + TiO2 layer (0.15 g, 60 V)	−0.282 ± 0.02	206.4 ± 18
		MT; CNTs layer + TiO2 layer (0.30 g, 50 V)	−0.217 ± 0.02	17.5 ± 5.2
		MT; CNTs layer + TiO2 layer (0.30 g, 60 V)	−0.439 ± 0.04	9.85 ± 0.8
		MT; CNTs layer	−0.233 ± 0.01	176.2 ± 22
D1 D2 D3 D4 D5 D6	Ti	Only MT	−0.015 ± 0.009	2.1 ± 0.069
		MT; LT	−0.011 ± 0.089	26.3 ± 1.979
	Ti6Al4V	Only MT	−0.225 ± 0.012	0.1 ± 0.009
		MT; LT	−0.022 ± 0.009	1.3 ± 0.067
	Ti13Nb13Zr	Only MT	−0.028 ± 0.007	0.1 ± 0.008
		MT; LT	−0.016 ± 0.006	44.7 ± 0.077
E1 E2 E3 E4	Ti13Zr13Nb	Only MT	−0.529 ± 0.042	150 ± 35
		MT; chitosan	−0.440 ± 0.022	256 ± 13
		MT; chitosan/Tween 20 coating	−0.517 ± 0.026	373 ± 19
		MT; chitosan/Tween 20/nanoPt coating	−0.239 ± 0.048	157 ± 20

Legend: MT mechanical treatment, MAO micro-arc oxidation, LT laser treatment.

**Table 5 materials-15-07556-t005:** Corrosion protection efficiency calculated for different surface modifications.

Procedure	Substrate	Surface Treatment	Corrosion Protection Efficiency (%)
A	Ti	No MT; AO	−631
		MT; Ag layer	−433
		MT; AO + Ag layer + chit/EU layer	−99
		MT; AO + Ag layer + chit/P4VP layer	−418
B	Ti13Nb13Zr	MT; MAO Ca + GP; 300 V	−73
		MT; MAO Ca + GP; 400 V	−221
		MT; MAO Ca + GP + Ag; 300 V	−107
		MT; MAO Ca + GP + Ag; 400 V	−192
C	Ti13Nb13Zr	MT; CNTs layer + TiO_2_ layer (0.15 g, 50 V)	+67
		MT; CNTs layer + TiO_2_ layer (0.15 g, 60 V)	−4790
		MT; CNTs layer + TiO_2_ layer (0.30 g, 50 V)	−315
		MT; CNTs layer + TiO_2_ layer (0.300 g, 60 V)	−133
		MT; CNTs layer	−4075
D	Ti	MT; LT	−1152
	Ti6Al4V	MT; LT	−1200
	Ti13Nb13Zr	MT; LT	−44,600
E	Ti13Nb13Zr	MT; chitosan	−71
		MT; chitosan/Tween 20 coating	−149
		MT; chitosan/Tween 20/nanoPt coating	−5

Legend: MT mechanical treatment, MAO micro-arc oxidation, LT laser treatment.

## Data Availability

Not applicable.

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
