# Peer review of "Influence of Surface Modification of Titanium and Its Alloys for Medical Implants on Their Corrosion Behavior"

_materials, 2022, doi:10.3390/ma15217556_

Round 1
Reviewer 1 Report
The article presented influence of surface modification of titanium and its alloys for medical implants on their corrosion behavior. The results are indeed useful for the community. I propose the acceptance of the article.
Author Response
Answers of authors to the comments of Reviewer No. 1
We would like to warmly thank the reviewer for his/her helpful comments and positive assessment of our paper.
Reviewer 2 Report
A very interesting manuscript. I recommend presenting Fig. 7(a) in the same manner as the other figures. The legends in Fig. 9, for all the representations, must have a bigger font size to be easy to read. Also, in Fig. 9., it is a bit confusing to understand just by using a) and b). Maybe reorganizing the entire Fig. 9 will make it better to understand.
Author Response
Answers of authors to the comments of Reviewer No. 2
In the beginning, we would like to warmly thank the reviewer for his/her helpful comments. They have contributed to the improvement of our manuscript.
- I recommend presenting Fig. 7(a) in the same manner as the other figures.
Answer: It has been done.
- The legends in Fig. 9, for all the representations, must have a bigger font size to be easy to read.
Answer: Figure 9 has been improved as suggested.
- Also, in Fig. 9., it is a bit confusing to understand just by using a) and b). Maybe reorganizing the entire Fig. 9 will make it better to understand
Answer: Figure 9 has been reorganized by a simple cutting of letters and better descriptions in the legend.
Reviewer 3 Report
This paper investigates the Influence of different surface modification of titanium and its alloys on their corrosion behavior. It is very useful for other researchers in this field. It can be accepted for publication after minor revision.
1. Table 4 should have the same format with the other tables.
2. Section 3 is two simple; the author should give more detailed description of the experimental results as shown in figure 1 to figure 10.
3. “In Figure 1, the nanotubular surfaces and slightly uniform other coatings are shown. “ In fact, as shown in figure 1, only figure 1(b) shows the tubular morphology.
4. Why different substrate were used? What is the effect of the subsrate on the formation process and microstructure of the surface layers?
Author Response
Answers of authors to the comments of Reviewer No. 3
In the beginning, we would like to warmly thank the reviewer for his/her helpful comments. They have contributed to the improvement of our manuscript.
- Table 4 should have the same format with the other tables.
Answer: The table has been modified as suggested.
- Section 3 is too simple; the author should give more detailed description of the experimental results as shown in figure 1 to figure 10.
Answer: The results have been joined with the discussion as suggested by the other reviewer, and the description has been enlarged.
- In Figure 1, the nanotubular surfaces and slightly uniform other coatings are shown. In fact, as shown in figure 1, only figure 1(b) shows the tubular morphology.
Answer: The description has been modified.
- Why different substrate were used? What is the effect of the substrate on the formation process and microstructure of the surface layers?
Answer: The detailed explanation has been inserted into the text.
Reviewer 4 Report
In the introduction part, the authors could give more detailed information for different oxides/coating preparation methods (anodic oxidation, micro-arc oxidation or hydrothermal oxidation), like their strength and drawbacks for medical implants.
Table 1, it’s better to unify the reference electrode for corrosion potential so that the readers can easily compare.
In table 3, why use four different chemical compositions of simulated body fluids for different corrosion tests? Please explain the specific conditions or use of each SRB in the tests.
The results descriptions are too short, even there is no descriptions for figures 5-10. The authors should combine the results and discussion parts together. Since for results part, there are ten figures and one table, but only 7 sentences in this part. At this condition, the readers can’t connect your discussions with the results showed above.
Author Response
Answers of authors to the comments of Reviewer No. 4
In the beginning, we would like to warmly thank the reviewer for his/her helpful comments. They have contributed to the improvement of our manuscript.
- In the introduction part, the authors could give more detailed information for different oxides/coating preparation methods (anodic oxidation, micro-arc oxidation or hydrothermal oxidation), like their strength and drawbacks for medical implants.
Answer: Appropriate information has been inserted into the introduction together with references.
- Table 1, it’s better to unify the reference electrode for corrosion potential so that the readers can easily compare.
Answer: All potentials are now related to the SCE.
- In table 3, why use four different chemical compositions of simulated body fluids for different corrosion tests? Please explain the specific conditions or use of each SRB in the tests.
Answer: The appropriate explanation has been inserted into the manuscript. Besides, the error in one for SBF compositions has been corrected and now three test solutions were shown to be used as in Table 3.
- The results descriptions are too short, even there is no descriptions for figures 5-10. The authors should combine the results and discussion parts together. Since for results part, there are ten figures and one table, but only 7 sentences in this part. At this condition, the readers can’t connect your discussions with the results showed above.
Answer: The results were joined with discussion and the description of results has been enriched as suggested also by the other reviewer.
Reviewer 5 Report
The main problem with the manuscript is that Tafel plots constructed from potentiodynamic polarization experiments are not a reliable way to measure corrosion rates of metals/alloys in a state of passivation.; despite the literature being full of erroneous examples where this has been done – see Table 1 in the manuscript.
The two plots in Figure 6 illustrate some of the problems with potentiodynamic based Tafel plots: The open-circuit potential in Figure 6a) are more positive than the Ecorr values determined from Figure 6b) and it likely that this difference would have increased if the OCP experiment had been allowed to confine for a longer period (days) as ennoblement is expected as the passive film stabilize; this can already be seen for Modification IV. The difference between OCP and Ecorr can be caused by a number of factors, such as thinning of the passive film during the period that the sample is held at extreme negative potential as well as capacitive charging affects. In the case where the passive thins this will regrow as the potential is swept in the positive direction, such that the anodic current flowing is mainly due to oxide growth (plus some capacitive charging), thus the current density determined at the so-called Ecorr has nothing to do with the corrosion rate. The true corrosion rate needs to be determined at the OCP, but even that is difficult as it takes days for the system to reach steady state and techniques like EIS yield an almost purely capacitive response. Unfortunately, there remains no reliable method for determining the true corrosion rate of metals/alloys in their passive state.
Nevertheless, it could be argued that the there is some usefulness in the large amount of data presented in the manuscript, as it does show that the majority of the various surface treatments that were designed to improve biomedical aspects of the implants do not result in unexceptionally high corrosion rates; although the technique used by the authors is not suitable for measuring the true corrosion rate, it is fair to say that the true corrosion current densities will be less than the values determined at Ecorr in the potentiodynamic Tafel plots. As such the manuscript could still have value after major revision.
Addition points that the Authors need to address include:
The manuscript needs a good proof reading, to improve both the English and the style. For example the first line of the abstract “Titanium and its alloys are often used for long-term implants to improve their surface properties, only in a small part to increase corrosion resistance.” makes no sense, probably part of the sentence is missing.
Lines 26 & 38: Titanium does not suffer from pitting or crevice corrosion in saline environments unless the temperature is high; far above that relevant to biomedical applications.
Line 27: Only currents can be referral to as anodic or cathodic, potentials are positise or negative with respect to a given reference electrode.
Table 1: a) All the corrosion potentials should be given against the same reference electrode, the use of multiple reference electrodes is confusing; b) The method used to determine the corrosion current densities should be stated; c) please indicate if the solution were deoxygenated or not; d) many of the values for the corrosion current densities in the table are clearly unrealistically high (even if they are in the literature), one would expect the corrosion rate of Ti in saline solution to be less than 1 micrometer per year, which is about 0.1 microamps/cm2, however some of the values quoted in the table are above 100 microamps/cm-2 which would be a corrosion rate in the region of 1 mm per year!
Line 197: A scan rate of 1 mV/s is too fast for a Tafel lot, especially for metals with a low corrosion rate as the capacitive charging currents can be of a similar magnitude to the corrosion current density.
Figure 1: It is clear from Figures 1b),c) and e) that the effective service areas of the samples is far larger than the geometric area; this also applies to Figures 2 & 3This means there is no real correlation between the currents measured in the potentiodynamic experiments and the corrosion rate. For example the very large surface area of the titania (i.e. oxide not metal) nanotubes will lead to very large capacitive charging currents flowing during the potentiodynamic measurements.
Figures 6 + others; The reference electrode should always be indicated on the potential axis.
Figure 7a): Open circuit potentials cannot be represented as bar charts, as the zero is meaningless, as it varies with the chosen reference electrode system.
Line 273: I would agree that 0.0003 nA/cm2 is unexpectedly low, but 374 nA/cm2 is actually higher than might be expected for the corrosion of Ti in a mild saline environment.
Line 293: The authors need to quote the value that they recommend for the “another limit” above which samples should be rejected.
Line 303: The corrosion potential moves to more positive potentials, not such thing as a more anodic potentials (see comment on line 27)
Line 315 to 334: Do the authors have any visual evidence to support their claims of pitting/crevice corrosion? The likelihood of such attack on Ti alloys in saline environments at 37C is extremely low.
Line 350: The porous SLM-made structures appear to have high corrosion rates because of the very high surface areas that areas that are exposed to the electrolyte. This issue has been known for many years as it also arises with porous Ti alloys made by more traditional methods such as powder compaction.
Author Response
Answers of authors to the comments of Reviewer No. 5
In the beginning, we would like to warmly thank the reviewer for his/her helpful comments. They have contributed to the improvement of our manuscript.
- The main problem with the manuscript is that Tafel plots constructed from potentiodynamic polarization experiments are not a reliable way to measure corrosion rates of metals/alloys in a state of passivation.; despite the literature being full of erroneous examples where this has been done – see Table 1 in the manuscript.
Answer: We agree that this method can bring out errors. Justifying our approach, there are only this technique, linear polarization method and long-term weight method commonly applied. The potentiodynamic method is here, and in all tests used to compare some surfaces and not to predict a real corrosion rate. We have inserted such an explanation, following the above statement, into the discussion.
- The two plots in Figure 6 illustrate some of the problems with potentiodynamic based Tafel plots: The open-circuit potential in Figure 6a) are more positive than the Ecorr values determined from Figure 6b) and it likely that this difference would have increased if the OCP experiment had been allowed to confine for a longer period (days) as ennoblement is expected as the passive film stabilize; this can already be seen for Modification IV. The difference between OCP and Ecorr can be caused by a number of factors, such as thinning of the passive film during the period that the sample is held at extreme negative potential as well as capacitive charging affects. In the case where the passive thins this will regrow as the potential is swept in the positive direction, such that the anodic current flowing is mainly due to oxide growth (plus some capacitive charging), thus the current density determined at the so-called Ecorr has nothing to do with the corrosion rate. The true corrosion rate needs to be determined at the OCP, but even that is difficult as it takes days for the system to reach steady state and techniques like EIS yield an almost purely capacitive response. Unfortunately, there remains no reliable method for determining the true corrosion rate of metals/alloys in their passive state.
Answer: We agree with this remark and now we do not consider the shift in corrosion potential as a measure of the tendency to corrosion rate change. We have inserted such an explanation, following the above statement, into the discussion.
- Nevertheless, it could be argued that the there is some usefulness in the large amount of data presented in the manuscript, as it does show that the majority of the various surface treatments that were designed to improve biomedical aspects of the implants do not result in unexceptionally high corrosion rates; although the technique used by the authors is not suitable for measuring the true corrosion rate, it is fair to say that the true corrosion current densities will be less than the values determined at Ecorr in the potentiodynamic Tafel plots. As such the manuscript could still have value after major revision.
Answer: We agree with this remark and inserted such an explanation, following the above statement, into the discussion.
- The manuscript needs a good proof reading, to improve both the English and the style. For example the first line of the abstract “Titanium and its alloys are often used for long-term implants to improve their surface properties, only in a small part to increase corrosion resistance.” makes no sense, probably part of the sentence is missing.
Answer: The manuscript has been verified by several of us and also by the Grammarly software. Despite that, the corresponding author has checked it once more and, regretfully, has found some grammar errors, including the above one.
- Lines 26 & 38: Titanium does not suffer from pitting or crevice corrosion in saline environments unless the temperature is high; far above that relevant to biomedical applications.
Answer: We agree with this remark to some extent. All passive metals and alloys can suffer from localized corrosion as the pH value sharply decreases in the crevices or cracks in the coating, and titanium below the value of 4 to 5 starts to dissolve. Following the above remark and understanding that our explanation has not been precise enough, we have changed our discussion on this point, based on two new references.
- Line 27: Only currents can be referral to as anodic or cathodic, potentials are positise or negative with respect to a given reference electrode.
Answer: It is right, we have made the change in the text.
- Table 1: a) All the corrosion potentials should be given against the same reference electrode, the use of multiple reference electrodes is confusing; b) The method used to determine the corrosion current densities should be stated; c) please indicate if the solution were deoxygenated or not; d) many of the values for the corrosion current densities in the table are clearly unrealistically high (even if they are in the literature), one would expect the corrosion rate of Ti in saline solution to be less than 1 micrometer per year, which is about 0.1 microamps/cm2, however some of the values quoted in the table are above 100 microamps/cm-2 which would be a corrosion rate in the region of 1 mm per year!
Answer: (1a) We have changed the table. (1b) The table is the only presentation of recent results and it is not a review paper. Therefore, we think that any interested reader can look at the literature reference and more details are unnecessary. (1c) See our answer above. (1d) We have only cited the values from different papers not discussing if they are true or not. We agree with a reviewer and, if we might prepare a review paper, we would certainly give the same remark.
- Line 197: A scan rate of 1 mV/s is too fast for a Tafel lot, especially for metals with a low corrosion rate as the capacitive charging currents can be of a similar magnitude to the corrosion current density.
Answer: We have chosen such potential change value based on a lot of previous research. In addition, the moderate effect of scan rate has been already shown (even if not for Ti) in, e.g. The effect of scan rate on the precision of determining corrosion current by Tafel extrapolation: A numerical study on the example of pure Cu in chloride containing medium. Diego A. Fischer, Ignacio T. Vargas, Gonzalo E. Pizarro, Francisco Armijo, Magdalena Walczak. Electrochimica Acta 313 (209) 457-467.
- Figure 1: It is clear from Figures 1b),c) and e) that the effective service areas of the samples is far larger than the geometric area; this also applies to Figures 2 & 3This means there is no real correlation between the currents measured in the potentiodynamic experiments and the corrosion rate. For example the very large surface area of the titania (i.e. oxide not metal) nanotubes will lead to very large capacitive charging currents flowing during the potentiodynamic measurements.
Answer: It is true and we have added such a comment in the discussion part.
- Figures 6 + others; The reference electrode should always be indicated on the potential axis.
Answer: It has been done in the revised manuscript.
- Figure 7a): Open circuit potentials cannot be represented as bar charts, as the zero is meaningless, as it varies with the chosen reference electrode system.
Answer: It has been modified.
- Line 273: I would agree that 0.0003 nA/cm2 is unexpectedly low, but 374 nA/cm2 is actually higher than might be expected for the corrosion of Ti in a mild saline environment.
Answer: We have made the text considering this remark.
- Line 293: The authors need to quote the value that they recommend for the “another limit” above which samples should be rejected.
Answer: We have made the text considering this remark.
- Line 303: The corrosion potential moves to more positive potentials, not such thing as a more anodic potentials (see comment on line 27)
Answer: We have made the text considering this remark.
- Line 315 to 334: Do the authors have any visual evidence to support their claims of pitting/crevice corrosion? The likelihood of such attack on Ti alloys in saline environments at 37C is extremely low.
Answer: We have made the text considering this remark. See our answer above.
- Line 350: The porous SLM-made structures appear to have high corrosion rates because of the very high surface areas that areas that are exposed to the electrolyte. This issue has been known for many years as it also arises with porous Ti alloys made by more traditional methods such as powder compaction.
Answer: We agree with this statement and have made the text considering this remark.
Reviewer 6 Report
1. The target of using different surface modification methods is not fully described in the introduction part.
2. Using different methods resulted in less characterization and deep analysis of the obtained results, which makes the paper to be like a technical report.
3. Most of the polarization curves did not show linear regions, sow how the authors calculated the values of icorr?. The cathodic and anodic Tafel slopes should be listed.
4. Table 4, corrosion protection efficiency (%) were listed in the Table. Are the values correct?
5. What is the phase composition of coatings?, For example, in which form Ag will be incorporated and how it would affect the properties?
6. The reasons for the selection of experimental conditions should be clearly described.
7. Some important references were not cited.
8. My suggestion is to focus on one method and provide more discussion and analysis. XRD, XPS, EIS are necessary.
In the present form, the manuscript can not be accepted for publication in Materials
Author Response
Answers of authors to the comments of Reviewer No. 6
In the beginning, we would like to warmly thank the reviewer for his/her helpful comments. They have contributed to the improvement of our manuscript.
- The target of using different surface modification methods is not fully described in the introduction part.
Answer: The explanation has been inserted into the introduction.
- Using different methods resulted in less characterization and deep analysis of the obtained results, which makes the paper to be like a technical report.
Answer: The explanation has been inserted into the introduction.
- Most of the polarization curves did not show linear regions, sow how the authors calculated the values of icorr?. The cathodic and anodic Tafel slopes should be listed.
Answer: The values of corrosion parameters have been determined by special software attached to the potentiostat.
- Table 4, corrosion protection efficiency (%) were listed in the Table. Are the values correct?
Answer: All values have been thoroughly checked again and, in several cases, corrected.
- What is the phase composition of coatings? For example, in which form Ag will be incorporated and how it would affect the properties?
Answer: The additional explanation has been inserted into the manuscript.
- The reasons for the selection of experimental conditions should be clearly described.
Answer: The additional explanation has been inserted into the manuscript.
- Some important references were not cited.
Answer: The cited references have been given based on a review of all relevant papers published in the last six years, present in Science Direct. It is impossible to make a good choice of the most important references, and if enlarging the base for the choice of references, e.g., for the last 20 years, the bibliography might be very huge. Therefore, now the introduction is focused to show the main directions of the most recent corrosion studies. We hope that the reviewer understands our attitude even if it is certain that we have not noticed all important achievements.
- My suggestion is to focus on one method and provide more discussion and analysis. XRD, XPS, EIS are necessary.
Answer: We have inserted all data already published which now gives a reader more information about chemical and phase composition for several studies. We have also mentioned that in some cases such investigations will be done, but the structure of coatings can be described. We hope that the reviewer understands our attitude even though we do not give any XRD, XPS, or EIS results in this paper. The key is to know about the compositions of coatings, important to understand corrosion degradation, and we think that we have achieved it.
Round 2
Reviewer 4 Report
All the corrections are Ok for me. The paper could be accepted in present form.
Author Response
All the corrections are Ok for me. The paper could be accepted in present form.
Answer: We are grateful for your final positive assessment of our research and the manuscript.
Reviewer 5 Report
The authors have greatly improved the manuscript and addressed most of my concerns. However, there are two points I still wish to raise.
Line 223: Impact of scan rate on Tafel plots: The numerical study referred to by the authors was on Cu and thus is not applicable to Ti alloys. The impact of scan rate becomes more important as the corrosion rate decreases, simply because the capacitive charging currents make up a larger fraction of the total measured current density. This is exasperated if the alloy is covered by a passive film or some coating. Remember for a capacitor current = C(dV/dt). However, as the authors only use their data for comparison purposes I am prepared to accept this.
Line 302: Why do the authors claim that titanium will start to dissolve below pH 4 to pH5? One can see from the Pourbaix diagram that this would only be the case under extremely reducing conditions, which are extremely unlikely in a biomedical environment; Titanium grade 2 does not undergo corrosion in seawater belwo 100 C. As for the new references added: the localized corrosion in refence 43 is due to the presence of fluoride, not chlorides; reference 44 is not reliable, as it only an extended abstract for a conference, and it would appear that the author is unaware that the majority of the anodic current density shown in his/her Fig 1 (at ca. pH 6) is due to oxide growth and dissolution of titanium. Furthermore, Ref 44 is a modelling study and does not report experimental corrosion rates; indeed it only predicted pH <4 for very tight crevices, less than 10 microns; which is well below the 25-100 microns of a typical crevice width found in corrosion works https://www.nrc.gov/docs/ML1122/ML11229A050.pdf and likely too tight for electrolyte to penetrate at ambient pressures. As such ref 44 should be removed.Overall, any claims for crevice/pitting corrosoion of Ti in biomedical conditions are at best highly dubious. The only possible exception may be highly porous specimens (powder metalurgy or 3d printed) where the crevice geometry could be extremely deep and narrow, but even here the excess corrosion reported is more likley to be due to the extremely large areas exposed to the electrolyte in such samples.
Line 301: Typo in the word “Here” at beginning of the sentence.
Author Response
- Line 223: Impact of scan rate on Tafel plots: The numerical study referred to by the authors was on Cu and thus is not applicable to Ti alloys. The impact of scan rate becomes more important as the corrosion rate decreases, simply because the capacitive charging currents make up a larger fraction of the total measured current density. This is exasperated if the alloy is covered by a passive film or some coating. Remember for a capacitor current = C(dV/dt). However, as the authors only use their data for comparison purposes I am prepared to accept this.
Answer: We agree with the reviewer`s comments and certainly will take into account in future research. In particular, we will check the effect of scan rate in planned experiments as it is very important.
Line 302: Why do the authors claim that titanium will start to dissolve below pH 4 to pH5? One can see from the Pourbaix diagram that this would only be the case under extremely reducing conditions, which are extremely unlikely in a biomedical environment; Titanium grade 2 does not undergo corrosion in seawater below 100 C.
Answer: We did not find such an assumption in the revised manuscript and we fully agree with the reviewer`s remark.
- As for the new references added: the localized corrosion in reference 43 is due to the presence of fluoride, not chlorides; reference 44 is not reliable, as it only an extended abstract for a conference, and it would appear that the author is unaware that the majority of the anodic current density shown in his/her Fig 1 (at ca. pH 6) is due to oxide growth and dissolution of titanium. Furthermore, Ref 44 is a modelling study and does not report experimental corrosion rates; indeed it only predicted pH <4 for very tight crevices, less than 10 microns; which is well below the 25-100 microns of a typical crevice width found in corrosion works. Overall, any claims for crevice/pitting corrosion of Ti in biomedical conditions are at best highly dubious. The only possible exception may be highly porous specimens (powder metallurgy or 3d printed) where the crevice geometry could be extremely deep and narrow, but even here the excess corrosion reported is more likely to be due to the extremely large areas exposed to the electrolyte in such samples.
Answer: We have agreed with the remark of the reviewer and deleted references 42-44, and the associated part of the manuscript.
- Line 301: Typo in the word “Here” at beginning of the sentence.
Answer: The error has been corrected.
Reviewer 6 Report
The answers provided by the authors to my questions related to polarization curves are not enough and are not correct. The questions remained unsolved. Since there are critical mistakes in the values of corrosion results and the authors did not pay any attention to check their results, I strongly recommend rejecting this paper.
Author Response
- The answers provided by the authors to my questions related to polarization curves are not enough and are not correct. The questions remained unsolved. Since there are critical mistakes in the values of corrosion results and the authors did not pay any attention to check their results, I strongly recommend rejecting this paper.
Answer: We acknowledge this response.